# Optimization of Laboratory Diagnostics of Primary Biliary Cholangitis: When Solid-Phase Assays and Immunofluorescence Combine

**DOI:** 10.3390/jcm11175238

**Published:** 2022-09-05

**Authors:** Federica Gaiani, Roberta Minerba, Alessandra Picanza, Annalisa Russo, Alessandra Melegari, Elena De Santis, Tommaso Trenti, Lucia Belloni, Silvia Peveri, Rosalia Aloe, Carlo Ferrari, Luigi Laghi, Gian Luigi de’Angelis, Chiara Bonaguri

**Affiliations:** 1Department of Medicine and Surgery, University of Parma, Via Gramsci 14, 43126 Parma, Italy; 2Gastroenterology and Endoscopy Unit, University Hospital of Parma, Via Gramsci 14, 43126 Parma, Italy; 3Laboratory of Clinical Chemistry and Hematology, University Hospital of Parma, Via Gramsci 14, 43126 Parma, Italy; 4Autoimmunity Unit, Department of Laboratory Medicine and Pathology, S. Agostino Estense Hospital, Via Giardini 1355, 41126 Baggiovara, Italy; 5Department of Laboratory Medicine and Pathology, S. Agostino Estense Hospital, Via Giardini 1355, 41126 Baggiovara, Italy; 6Unit of Clinical Immunology, Allergy and Advanced Biotechnologies, Azienda Unità Sanitaria Locale—IRCCS of Reggio-Emilia, Viale Risorgimento 80, 42123 Reggio-Emilia, Italy; 7Allergology Unit, Guglielmo da Saliceto Hospital, Via Giuseppe Taverna 49, 29121 Piacenza, Italy; 8Laboratory of Viral Immunopathology, Unit of Infectious Diseases and Hepatology, University Hospital of Parma, Via Gramsci 14, 43126 Parma, Italy

**Keywords:** primary biliary cholangitis, anti-mitochondrial antibodies, indirect immunofluorescence, solid-phase assay

## Abstract

The laboratory diagnostics of primary biliary cholangitis (PBC) have substantially improved, thanks to innovative analytical opportunities, such as enzyme-linked immunosorbent assays (ELISA) and multiple immunodot liver profile tests, based on recombinant or purified antigens. This study aimed to identify the best diagnostic test combination to optimize PBC diagnosis. Between January 2014 and March 2017, 164 PBC patients were recruited at the hospitals of Parma, Modena, Reggio-Emilia, and Piacenza. Antinuclear antibodies (ANA) and anti-mitochondrial antibodies (AMA) were assayed by indirect immunofluorescence (IIF), ELISA, and immunodot assays (PBC Screen, MIT3, M2, gp210, and sp100). AMA-IIF resulted in 89.6% positive cases. Using multiple immunodot liver profiles, AMA-M2 sensitivity was 94.5%, while anti-gp210 and anti-sp100 antibodies were positive in 16.5% and 17.7% of patients, respectively. PBC screening yielded positive results in 94.5% of cases; MIT3, sp100, and gp210 were detected by individual ELISA test in 89.0%, 17.1%, and 18.9% of patients, respectively. The association of PBC screening with IIF-AMA improved the diagnostic sensitivity from 89.6% to 98.2% (*p* < 0.01). When multiple immunodot liver profile testing was integrated with AMA-IIF, the diagnostic sensitivity increased from 89.1% to 98.8% (*p* < 0.01). The combination of IIF with solid-phase methods significantly improved diagnostic efficacy in PBC patients.

## 1. Introduction

Primary biliary cholangitis (PBC) is a chronic and often progressive cholestatic liver disease, characterized by the autoimmune destruction of the intrahepatic bile ducts [1]. An increased value of specific serum anti-mitochondrial antibodies (AMA) [2] is the hallmark of the disease, accompanied by the evidence of cholestasis. In accordance with the largest English epidemiological study, PBC is a rare disease, more represented in women than in men, with a prevalence of about 35/100.000 and an annual incidence of 2–3/100.000 [3,4]. The disease is mostly represented in late adulthood, with 65 years as the mean age at diagnosis [1].

The etiology of PBC has not yet been clarified, but is thought to be multifactorial due to a combination of environmental and genetic risk factors [5].

The diagnosis is confirmed by the evidence of sustained (>6 months) elevated values of alkaline phosphatase (ALP), accompanied by positive serum AMA at a titer > 1/40, and/or to specific antinuclear antibodies (ANA) [2].

According to the European Association for the Study of the Liver (EASL) guidelines [2], at least two of the following three criteria must be met for confirming a diagnosis of PBC: (1) serum AMA and/or ANA positivity, (2) cholestatic pattern of liver biochemistry tests with at least one increased value among serum bilirubin, ALP, or gamma-glutamyltransferase (GGT), or (3) diagnostic liver histology.

AMA positivity can be considered as the serological hallmark of PBC, being typically observed in over 90% of patients. Evidence of autoantibodies by indirect immunofluorescence (IIF) or enzyme-linked immunosorbent assays (ELISA) is highly specific for this condition. Although AMA positivity is a strong indicator of PBC in patients with otherwise unexplained cholestasis, AMA reactivity is only sufficient for diagnosing PBC in combination with abnormal values from these tests [5,6,7,8,9].

Besides AMA, ANA are present in about 50–70% of PBC patients, with speckled multiple nuclear dot (MND), rim-like/membranous (RL/M), and anti-centromere (ACA) as the most frequent patterns observed [10]. Nearly 30–50% of PBC patients show positivity for MND (i.e., anti-sp100 reactivity) and RL/M (i.e., anti-gp210 reactivity), thus reflecting a low diagnostic sensitivity [9,11,12]. As AMA-negative PBC patients are at risk of misclassification, the detection of PBC-specific ANA can sustain diagnosis [7].

While AMA, MND, and RL/M were primarily detected as immunofluorescent patterns, the identification of target autoantigens, such as the E2 component of pyruvate dehydrogenase (PDC-E2), sp100, and gp210, has allowed for the development of ELISA assays and specific immunodot tests based on recombinant or purified antigens [13,14,15,16,17]. Increased awareness of the serological associations of PBC, along with the widespread use of blood test-based screening in the community, have significantly changed the initial presentation of PBC patients in recent years, so that patients diagnosed with clinically overt disease (e.g., advanced liver disease) can now be identified earlier using abnormal liver serum tests at screening [2].

The clinical course of PBC is variable, may lead to hepatic fibrosis, and ultimately, to hepatic cirrhosis or hepatic failure, with enhanced risk of evolution towards hepatic cellular carcinoma [2,18]. Because of their high sensitivity and specificity, AMA targeting the PDC-E2 are currently considered as a diagnostic hallmark of PBC [19], although their positivity and titer are poor predictors of outcome [9].

Current evidence suggests that no established immunological marker can efficiently predict the progression towards end-stage liver cirrhosis. The available prognostic models have been developed based on clinical and biochemical variables (especially bilirubin) and have been tested in patients with advanced disease [20]. Therefore, they are virtually unsuitable for the prognostication of patients with early disease. Hence, new prognostic biomarkers would be needed for early diagnosis and for tailoring follow-up treatments, according to patients’ characteristics.

Evidence has been provided that PBC-specific ANA, particularly anti-gp210 or RL/M, may be associated with poor prognosis and more aggressive disease [9]. Although its clinical impact remains uncertain, the assessment of ANA in PBC patients seems promising. This unmet clinical need has prompted us to investigate whether PBC-specific ANA (i.e., anti-gp210 and anti-sp100 antibodies) may provide meaningful diagnostic and prognostic data. The study aimed to explore some substantial issues in PBC diagnostics. In particular, we investigated whether (a) the identification of PBC-specific autoantibodies against gp210 and sp100 would increase the diagnostic sensitivity of immunological testing for PBC, (b) the identification of AMA and PBC-specific anti-gp210 and anti-sp100 by immunodot and ELISA assays (based on molecularly defined antigens) would improve the sensitivity when compared to immunofluorescence-based techniques, and (c) the adoption of panels of autoantibodies would allow for the diagnosis of PBC in AMA negative patients, with the aim to minimize the risk of misclassification.

Our research strategy included the integration and review of previously published data, and the presentation of preliminary and partial results of the complete multicenter study is entirely illustrated in the present paper [21].

## 2. Materials and Methods

### 2.1. Study Design

A multicenter study was carried out at the hospitals of Parma, Modena, Reggio Emilia, and Piacenza between January 2014 and March 2017, recruiting patients diagnosed with PBC or with suspected PBC, according to the guidelines provided in [2] (Figure 1). 

These referral centers were identified for their vast experience with AMA-IIF testing and for the high number of tests performed yearly (i.e., in 2014, 978 tests were performed in Parma, 999 in Reggio-Emilia, 1087 in Modena, and 1638 in Piacenza, respectively). A signed informed consent was collected from all the recruited patients.

At the time of enrollment, a blood sample was taken from each patient, for a total of 12 mL. Serum was separated by conventional centrifugation procedures and stored at −80 °C until testing. All measurements were centralized in the laboratory of clinical chemistry and hematology of the University Hospital of Parma. In this laboratory, all samples were examined by using different available approaches, including immunofluorescence and solid-phase methods (i.e., ELISA and immunoblotting). The results of antibody testing with different analytical methods and for different target autoantigens located in distinct subnuclear structures were assessed. Clinical data including comorbidities and the histological findings of liver biopsies, when available, were also collected at enrollment and were then correlated with the autoantibody test results.

### 2.2. Laboratory Assays

Indirect immunofluorescence (Alphadia, Wavrem, Belgium, provided by Alifax, Padova, Italy) was applied on Hep-2 cells and on rat kidney, stomach, and liver sections, respectively, for assessing ANA and AMA. The initial dilution was 1:80, according to the manufacturer’s instructions, and the slides were then reviewed by two skilled laboratory professionals. All tests were performed with Multiple Immunodot Liver profile 7 Ag^®^ (Alphadia, Wavrem, Belgium, Alifax, Padova, Italy), according to the manufacturer’s instruction. This immunodot contained the PBC-associated antigens M2/native PDC (E1, E2, E3 subunits of Pyruvate Dehydrogenase Complex, purified from bovine heart), gp120 (recombinant human), and sp100 (recombinant human), as well as the autoimmune hepatitis specific antigens LKM1 (recombinant human), LC1 (recombinant human), SLA (purified from rat liver), and F-actin (purified from rabbit skeletal muscle). Four ELISA kits were used (Inova, San Diego, CA, USA): QUANTA Lite^®^ PBC screen IgG/IgA, QUANTA Lite^®^ (MIT3), QUANTA Lite^®^ gp210, and QUANTA Lite^®^ sp100. PBC Screen IgG/IgA ELISA assay contained purified recombinant antigen MIT3 (immunodominant portion of PDC-E2, BCOAD-E2, OGDC-E2) and a purified fragment of gp210 and sp100. 

All the collected serum samples were tested for ANA and AMA, PBC Screen, MIT3, M2, gp210, and sp100 antigens.

### 2.3. Statistical Analysis

The diagnostic performance in terms of agreement among different assays (i.e., IIF, ELISA, immunodot) was defined for each PBC-associated autoantibody by means of Cohen’s kappa, with 95% confidence interval (95% CI). The agreement was rated accordingly to Altman et al. [22]. Differences between PBC-specific autoantibody values (positive or negative) in different histological grades were assessed with χ^2^ test. The statistical analysis was performed with MedCalc Statistical Software version 20.114, MedCalc Software Ltd, Ostend, Belgium. 

## 3. Results

Throughout the study period, a total of 176 serum samples were collected from the same number of patients with established or suspected PBC. In 12 out of 176 patients (6.8%), the diagnosis of PBC could not be confirmed; therefore, they were excluded from further analysis (Figure 1). Overall, 3 of these 12 patients were diagnosed with autoimmune hepatitis, 1 with primary sclerosing cholangitis, and 8 with HCV-related hepatitis. The final study population consisted of 164 patients with a confirmed diagnosis of PBC (mean age, 63.5 years; range 34–89 years; male/female ratio, 1:9), in whom all autoantibody profiles were evaluated (Figure 1). The clinical characteristics of the enrolled patients are shown in Table 1. 

### 3.1. Detection of Autoantibodies with Different Analytical Techniques

Overall, 147 out of 164 (89.6%) PBC patients using the IIF assay with stomach/kidney/liver tissue were positive for AMA. Indirect immunofluorescence revealed positivity for Hep2 cells in 24 out of 164 (14.6%), and 25 out of 164 (15.2%) PBC patients were positive for MND patterns and RL/M, respectively. The multiple immunodot liver profile revealed a 94.5% sensitivity for AMA-M2, whereas anti-sp100 and anti-gp210 antibodies were positive in 17.7% and 16.5% patients, respectively. The ELISA PBC screen was positive in 155 out of 164 (94.5%) PBC patients; moreover, 146 (89.0%), 28 (17.1%), and 31 (18.9%) patients, respectively, tested positive for MIT3, sp100, and gp210 using specific ELISA tests. The autoantibody profiles of the 164 PBC patients are displayed in Table 1.

### 3.2. Agreement among Different Assays for PBC-Specific Autoantibodies

The results for AMA, AMA-M2, MIT3, and IIF-AMA showed a moderate agreement (kappa values between 0.538 and 0.465), while a more satisfactory agreement was detected between AMA-M2 and MIT3 (kappa value, 0.698) (Table 2). A noteworthy agreement was found for PCB-specific ANA between anti-sp100dot and anti-sp100 ELISA and between anti-gp210 dot and anti-gp210 ELISA (kappa values, 0.875), while a lower agreement was observed between MND ANA pattern and anti-sp100 ELISA or anti-sp100dot (kappa values of 0.718 and 0.636, respectively). A satisfactory agreement was also found between RL/M ANA pattern and anti-gp210 immunodot or anti-gp210 ELISA (kappa values of 0.735 and 0.718) (Table 2).

### 3.3. Overlap of PBC Specific Autoantibodies in PBC Samples

Regarding ELISA assays (i.e., MIT3, sp100, and gp210), among all patients, 22 (13.4%) showed double reactivity for MIT3 and sp100; 20 (12.2%) for MIT3 and gp210; and 4 (2.5%) displayed reactivity for all these antigens, while no patient displayed combined reactivity for both sp100 and gp210. Regarding the multiple immunodot liver profile, 23 of all cases (14.0%) displayed double reactivity for AMA-M2 and sp100; 16 (9.8%) for AMA-M2 and gp210; and 5 (3.1%) for all antigens, while none of them displayed combined positivity for sp100 and gp210. The results of the ELISA tests and the multiple immunodot liver profile for PBC-specific autoantibodies are shown in Figure 2a,b, respectively.

### 3.4. Combined Diagnostic Value of PBC Specific Autoantibodies

By combining the PBC screen assay with IIF-AMA, the diagnostic sensitivity significantly increased from 89.6% to 98.2% (*p* < 0.01). Likewise, the combination of the multiple immunodot liver profile and IIF-AMA increased the diagnostic sensitivity from 89.6% to 98.8% (*p* < 0.01). The positivity for single or combined ELISA and for the multiple immunodot liver profile in IIF-AMA negative patients is shown in Table 3.

A proposed flowchart for PBC diagnosis based on the availability of the PBC screen or liver dot profile is shown in Figure 3a,b.

### 3.5. Prognostic Value of PBC-Specific ANA

Histological staging was available for 50 PBC patients, among whom 29 were graded according to the Batts–Ludwig scale [23]. The ability of PBC-specific markers to identify patients presenting with severe disease was hence tested by comparing the positivity of nuclear autoantibodies (anti-sp100 and anti gp-120) with the histological classification (i.e., early versus advanced disease). The percentage of positivity for anti-sp100 and/or anti-gp120 antibodies depended on stage. The overall positivity percentage was 15.4% vs. 50% in low grade vs. high grade, respectively (grade > 1), *p* = 0.05. The aim of correlating the severity of the disease with antibody positivity was to propose the possibility of the early determination of the risk of developing severe disease.

## 4. Discussion

Laboratory tests for the research of specific autoantibodies, especially AMA, play a crucial role in PBC diagnosis, making the reliability of liver autoimmune serology of paramount importance in this clinical setting [2,9,24,25,26]. The screening for PBC-related autoantibodies (i.e., AMA and ANA) is conventionally performed, especially in European countries, using IIF instead of solid-phase test systems (i.e., ELISA and immunodot assays). IIF has many shortcomings, such as high inter-observer variability, the need for trained laboratory staff, its lack of suitability for full automation, and a poor degree of international standardization. The interpretation of IIF pattern can also be challenging due to the physiopathology of PBC. In particular, AMA IIF may be confused with other cytoplasmic antibodies not directly associated with PBC, such as anti-cardiolipin antibodies [26]. Notably, some antinuclear antibodies frequently encountered in patients with rheumatic diseases (e.g., anti-centromere and speckled) can coexist with PBC-specific ANA antibodies, contributing to generate a controversial fluorescent pattern [27]. Indeed, almost half of the patients enrolled in the present study are affected by other autoimmune diseases (Table 1), underlining how frequently the fluorescent pattern may be misleading.

The new solid-phase ELISA and immunodot tests, especially those using recombinant proteins such as MIT-3, now appear to be significantly more sensitive than IIF [11]. Furthermore, the detection of antibodies against gp210 and sp100 by molecular testing is more accurate than that using IIF on Hep2 cells [9].

The results of this multicenter study show that the PBC screen had an overall diagnostic sensitivity of 94.5% in a cohort of ascertained PBC patients, 10.4% of whom were AMA negative with IIF testing. In particular, the PBC screen exhibited a satisfactory diagnostic performance for patients without detectable IIF AMA (14/17, 64.7%) (Table 3), confirming and corroborating previous evidence published by Liu et al. [28]. Unlike, the M2/PBC immunodot test displayed an overall sensitivity of 94.5% (Table 1), but it was only capable to identify 9 out of 17 PBC patients without detectable IIF AMA (Table 3).

The better diagnostic sensitivity of the PBC screen observed in this study can be explained by the capability of this technique to also detect anti-gp210 and anti-sp100 antibodies, as previously reported by Liu et al. [28]. Notably, among all PBC patients, sp100 was detected by ELISA in 17.1% of cases and by immunodot in 17.7%; regarding gp210, ELISA returned positive results in 18.9% and immunodot in 16.5% of cases, respectively (Table 1).

Our findings also confirm the significant overlap of all PBC specific autoantibodies, which was found in 46 out of 164 (28%) patients using ELISA and in 44 out of 164 (26.8%) patients using immunodot.

As expected, the comparison of different techniques (i.e., IIF, ELISA, immunodot) for the identification of AMA and ANA PBC-specific antibodies revealed that ELISA and immunodot displayed the highest concordance (Table 2).

Although no reference technique is currently available for diagnosing PBC, our findings suggest that the risk of misdiagnosing AMA-negative PBC patients significantly decreased using innovative techniques such as ELISA or immunodot based on recombinant antigens, either alone or in combination with IIF. As recently reported for autoimmune rheumatic diseases [29], the combination of solid-phase techniques with AMA IIF may be useful to enhance sensitivity from 89.6% to 98.2% and 98.8% when combined with ELISA and immunodot, respectively. These findings suggest that IIF AMA should not be used alone as a first-line assay, but in combination with a new solid-phase technique (e.g., ELISA, PBC screen, or immunodot) to increase its accuracy for diagnosing PBC.

The concordance rates between ELISA and immunodot corroborate the advisability of combining AMA IIF with either one of the two solid-phase techniques. Many factors influence the choice of the best option, including expertise, available laboratory technologies, and economic resources. Therefore, we suggest the use of the PBC screen in association with AMA as first-line investigation for diagnosing PBC, especially when the pre-test probability is low. In this case, the possibility to perform single ELISA tests in sequence, combined with IIF, with the aim to find autoantibodies against each antigen, could allow for a precise diagnosis, limiting costs of the complete panel offered by the multiple immunodot liver profile. On the other hand, in specialized referral laboratories where the prevalence of patients needing PBC-specific antibodies evaluation is assumed to be higher, the option of contextually assessing all the relevant PBC-specific antibodies could lead to a preference for the use of multiple immunodot liver profile as a first-line assay, in association with AMA IIF. Typically, only referral laboratories for the study of autoimmunity possess the availability of both solid-phase tests and immunofluorescence, the proposed diagnostic flowchart aims to advise the optimization of PBC diagnostic to avoid potentially unproductive and expensive tests, especially for smaller laboratories.

With regards to the prognostic value of antibody tests, the findings of the present study suggest that the positivity of autoantibodies anti-sp100 and anti-gp120 could be related to the higher grade of histological severity, in agreement with the guidelines of the British Society of Gastroenterology [9], although this has not been confirmed by other studies [24,25]. Although these data are too limited to be adopted as a standard method for the determination of the prognosis of PBC, a complete characterization of specific autoantibodies in PBC patients could help validate their prognostic value in wider cohorts.

The first limitation of the study is represented by the number of patients enrolled, which is limited for drawing conclusions that can be applied to the general population. At the same time, it can be deemed remarkable in regards to PBC epidemiology, since the results were derived from the enrollment in only 4 centers, located in a small region. Second, the small percentage of available liver histology is another weakness of the study, as it was not proposed to all the patients enrolled, since it was not strictly necessary to obtain the diagnosis. Therefore, it would have been unethical to have all subjects undergo a potentially risky and invasive procedure for the research purposes only.

On the other hand, the standardization of laboratory procedures and the rigorous collaboration among the research groups in the 4 centers represents a strength of the project, guaranteeing both the reliability of antibody determination and the eventual adoption of different laboratory diagnostic protocols derived from the results of the multicenter study.

Innovative technological and analytical opportunities have allowed for substantial improvements in laboratory diagnostics in the field of PBC. In this new scenario, the clinical governance of the autoimmune diagnostics of liver diseases is crucial [14]. 

Tests performance in the diagnostic pathway of PBC should always follow an algorithm planned with the hepatologist. As the first step for the study of liver autoimmunity is represented by AMA and ANA determination by IIF, an “AMA reflex” profile could be suggested, with the association of ELISA and immunodot, to minimize misdiagnosis and obtain potentially prognostic data.

Large prospective population studies are required to validate the diagnostic efficacy and cost/benefit ratio of the combination of solid-phase methods (i.e., ELISA or immunodot) with AMA-IIF in the diagnostic approach to patients with PBC.

## Figures and Tables

**Figure 1 jcm-11-05238-f001:**
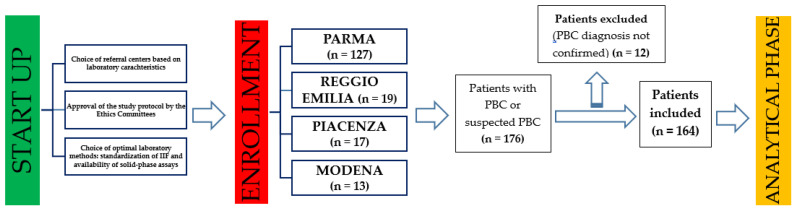
Structure of the study and enrollment process. IIF: indirect immunofluorescence; PBC: primary biliary cholangitis.

**Figure 2 jcm-11-05238-f002:**
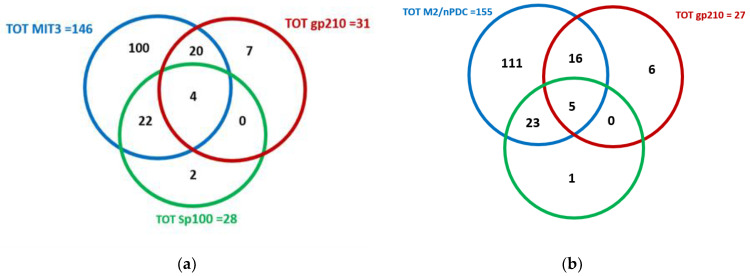
Overlap of ELISA tests (**a**) and multiple immunodot liver profile (**b**) for PBC-specific autoantibodies. ELISA: enzyme-linked immunosorbent assay; PBC: primary biliary cholangitis.

**Figure 3 jcm-11-05238-f003:**
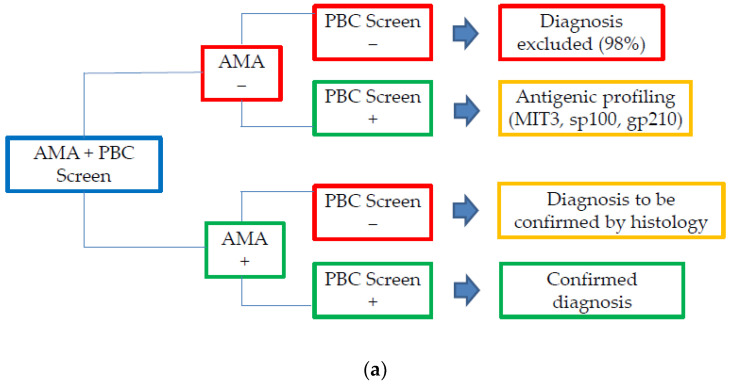
A proposed flowchart for PBC diagnosis based on the availability of the PBC screen (**a**) or liver dot profile (**b**).

**Table 1 jcm-11-05238-t001:** Clinical characteristics of the enrolled patients (n = 164).

	n° Patients (%)
Sex:	
- M	21 (12.8%)
- F	143 (87.2%)
Age (mean (range))	63.5 (34–89)
Center of enrollment:	
- Parma	115 (70.1%)
- Reggio Emilia	19 (11.6%)
- Piacenza	17 (10.4%)
- Modena	13 (7.9%)
ALP at diagnosis (mean (range))	44–1341 (211.3)
n/a	29
Other autoimmune diseases:	
- Yes	73 (44.5%)
- No	91 (55.5%)
Comorbidities:	
- autoimmune thyroiditis	20 (27.4%)
- autoimmune hepatitis	17 (23.3%)
- Sjogren Syndrome	15 (20.6%)
- scleroderma	11 (15.0%)
- Raynaud Syndrome	11 (15.0%)
- celiac disease	2 (2.7%)
- psoriasis	4 (5.5%)
- autoimmune diabetes	4 (5.5%)
- rheumatoid arthritis	3 (4.1%)
- psoriatic arthritis	2 (2.7%)
- rheumatic polymyalgia	3 (4.1%)
- IBD	3 (4.1%)
- SLE	5 (6.8%)
- glomerulopathy	1 (1.4%)
- polyarthritis	1 (1.4%)
- undifferentiated connectivitis	2 (2.7%)
Histology (hepatic biopsy):	
- Yes	50 (30.5%)
- No	114 (69.5%)
AMA IIF positive	147 (89.6%)
Liver Dot:	
- M2	155 (94.5%)
- gp210	27 (16.5%)
- sp100	29 (17.7%)
ELISA:	
- PBC screen	155 (94.5%)
- MIT3	146 (89.0%)
- gp210	31 (18.9%)
- sp100	28 (17.1%)
ANA:	
- ND	24 (14.6%)
- NL	25 (15.2%)
- Centromere	23 (14.0%)
- Speckled	17 (10.4%)
- Nucleolar	2 (1.2%)
- Homogeneous	3 (1.8%)

ALP: alkaline phosphatase; IBD: inflammatory bowel disease; SLE: systemic lupus erythematosus; AMA IIF: anti-mitochondrial antibody indirect immunofluorescence; ELISA: enzyme-linked immunosorbent assay; PBC: primary biliary cholangitis; ANA: anti-nuclear antibody.

**Table 2 jcm-11-05238-t002:** Agreement among results obtained using the IIF methods, multiple immunodot liver profile, and ELISA.

	Overall Agreement	Cohen’s Kappa (95% CI)
** *Anti-mitochondrial Ab* **		
AMA-IIF vs. M2/nPBC	0.921	0.538 (0.416–0.660)
AMA-IIF vs. MIT3	0.891	0.465 (0.347–0.583)
M2/nPBC vs. MIT3	0.945	0.694 (0.599–0.797)
** *Anti-sp100 Ab* **		
MND-IIF vs. sp100-dot	0.903	0.636 (0.550–0.722)
MND-IIF vs. sp100-ELISA	0.927	0.718 (0.640–0.796)
sp100-dot vs. sp100-ELISA	0.964	0.875 (0.825–0.925)
** *Anti-gp210 Ab* **		
RL/M-IIF vs. gp210-dot	0.927	0.717 (0.638–0.796)
RL/M-IIF vs. gp210-ELISA	0.927	0.735 (0.661–0.809)
gp210-dot vs. gp210-ELISA	0.964	0.875 (0.828–0.928)

AMA IIF: anti-mitochondrial antibody indirect immunofluorescence; PBC: primary biliary cholangitis; MND: multiple nuclear dot; ELISA: enzyme-linked immunosorbent assay; RL/M: rim-like/membranous.

**Table 3 jcm-11-05238-t003:** Combined diagnostic value of PBC-specific autoantibodies positivity in AMA-IIF negative PBC patients (n° = 17). ELISA: enzyme-linked immunosorbent assay; PBC: primary biliary cholangitis; PDC: pyruvate dehydrogenase complex.

	n° Patients (%)
** *ELISA* **	
PBC + MIT3	5 (29%)
PBC + gp210	4 (24%)
PBC + sp100	1 (6%)
PBC + MIT3 + sp100	2 (12%)
PBC + MIT3 + sp100 + gp210	1 (6%)
gp210	1 (6%)
negative	3 (17%)
** *Multiple Immunodot Liver Profile* **	
M2/nPDC	6 (35%)
gp210	5 (29%)
sp100	1 (6%)
M2/nPDC + sp100	2 (12%)
M2/nPDC + sp100 + gp210	1 (6%)
negative	2 (12%)

## Data Availability

All available data are published in the present article.

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
