# Peer review of "Optimization of Laboratory Diagnostics of Primary Biliary Cholangitis: When Solid-Phase Assays and Immunofluorescence Combine"

_jcm, 2022, doi:10.3390/jcm11175238_

Round 1

Reviewer 1 Report

AMA-M2 positivity seems to be a necessary and sufficient condition for the diagnosis of PBC. Does the diagnostic ability of PBC change when AMA-M2 is used in screening tests compared to other tests?

Author Response

Thank you very much for your comments, you can fing here attached the answer.

Reviewer 2 Report

Dear Authors,

Thank you for your work but I have found a previous publication of your work cited below , that matches the results of this manuscript (only one patient is removed from the original publication to change the number from 165 to 164 patients. Could you kindly explain?

Bonaguri C, Melegari A, Picanza A, Russo A, De Santis E, Trenti T, Parmeggiani M, Belloni L, Savi E, de'Angelis GL, Gaiani F, Ferrari C, Lippi G. Association of solid-phase assays to the indirect immunofluorescence in primary biliary cholangitis diagnosis: Results of an Italian multicenter study. Autoimmun Rev. 2019 Nov;18(11):102389. doi: 10.1016/j.autrev.2019.102389. Epub 2019 Sep 11. PMID: 31520799.

Author Response

(The authors gave the same response as above.)

Reviewer 3 Report

The authors have compared the performance of serological tests, single or in combinations, used to diagnose PBC. The topic is of interest to hepatologists. I have the following minor comments:

Methods

Figure 1: the left part of the figure (Start up) suits in a project description but does not seem relevant for the scientific content of the manuscript. I.e. The chose of optimal laboratory methods is not described in the manuscript. The number of patients enrolled from each center is also mentioned in the Results section in Table 1. The lines in the Table 1 are displaced and the layout should be adjusted in the final version.

Line 118-119 suggests that the number of tests run at each center has been similar (978 to 1638/year in 2014) whereas the number of recruited patients differed by a factor of ten (13 vs 127 patients included). Was there any particular reason that the number of patients / number of tests-ratio differed so much between centers and could this patient selection affect the study?

Line 132 and onwards: Please quote producer (company, city, country) and product code for the assays used in the study. The company providing them in Italy does not seem relevant?

Line 151: please define CBP or replace with PBC?

Methods and Results

Liver biopsy was available for 50 of 164 patients. Was there a gold standard for the PBC diagnosis used for the remaining 114 patients in the study?  

3.5 Prognostic value. The antibodies measured at one time-point must be referred to as associated with histological stage. The study design did not allow for “identifying patients at increased risk of developing severe disease”. The patients already had a certain disease severity at the time of inclusion. Please rephrase.

Discussion:

The last two paragraphs of the Discussion are well written. Please include a “strengths and limitations” paragraph above, for instance after line 325. This will highlight how the manuscripts contributes.

Author Response

Thank you very much for your comments, you can fing here attached the answers.

Round 2

Reviewer 1 Report

This paper has been properly revised.

Author Response

Thenk you very much for your effort in reviewing our paper. I've greatly appreciated your constructive comments, which have allowed to improve the manuscript.

Reviewer 2 Report

I would like to thank the authors for their explanation, I would recommend they add their explanation of a previous publication in the introduction section.

Author Response

Thank you very much for the suggestion, a specific sentence was added in the introduction, to explain the meaning of previously published data, compared to the present paper.